# Supplementation with *Spirulina platensis* Prevents Damage to Rat Erections in a Model of Erectile Dysfunction Promoted by Hypercaloric Diet-Induced Obesity

**DOI:** 10.3390/md20080467

**Published:** 2022-07-22

**Authors:** Iara Leão Luna de Souza, Bárbara Cavalcanti Barros, Elba dos Santos Ferreira, Fernando Ramos Queiroga, Luiz Henrique César Vasconcelos, Lydiane de Lima Tavares Toscano, Alexandre Sérgio Silva, Patrícia Mirella da Silva, Fabiana de Andrade Cavalcante, Bagnólia Araújo da Silva

**Affiliations:** 1Departamento de Ciências Biológicas e Saúde, Universidade Estadual de Roraima, Boa Vista 69306-530, RR, Brazil; 2Curso de Farmácia, Centro Universitário Estácio da Amazônia, Boa Vista 69306-530, RR, Brazil; 3Programa de Pós-graduação em Produtos Naturais e Sintéticos Bioativos, Centro de Ciências da Saúde, Universidade Federal da Paraíba, João Pessoa 58051-900, PB, Brazil; barbaracavalcante@ltf.ufpb.br (B.C.B.); elbaferreira@ltf.ufpb.br (E.d.S.F.); fernandoqueiroga@ltf.ufpb.br (F.R.Q.); lhcv@academico.ufpb.br (L.H.C.V.); fabianacavalcante@ltf.ufpb.br (F.d.A.C.); bagnolia@ltf.ufpb.br (B.A.d.S.); 4Centro de Ciências da Saúde, Departamento de Fisiologia e Patologia, Universidade Federal da Paraíba, João Pessoa 58051-900, PB, Brazil; 5Centro de Ciências da Saúde, Departamento de Educação Física, Universidade Federal da Paraíba, João Pessoa 58051-900, PB, Brazil; lyditavares@hotmail.com (L.d.L.T.T.); alexandresergiosilva@yahoo.com.br (A.S.S.); 6Programa de Pós-graduação em Biologia Celular e Molecular, Centro de Ciências Exatas e da Natureza, Universidade Federal da Paraíba, João Pessoa 58051-900, PB, Brazil; p.mirella.dasilva@dbm.ufpb.br; 7Centro de Ciências da Saúde, Departamento de Ciências Farmacêuticas, Universidade Federal da Paraíba, João Pessoa 58051-900, PB, Brazil

**Keywords:** *Spirulina platensis*, algae, erectile dysfunction, adiposity, antioxidant effect

## Abstract

Erectile dysfunction (ED) is the inability to achieve and/or maintain a penile erection sufficient for sexual satisfaction. Currently, many patients do not respond to the pharmacotherapy. The effects of a supplementation with *Spirulina platensis*, were evaluated in a model of ED induced by hypercaloric diet consumption. Wistar rats were divided into groups fed with standard diet (SD) or hypercaloric diet (HD) and supplemented with this alga at doses of 25, 50 or 100 mg/kg. Experimental adiposity parameters and erectile function were analyzed. In SD groups, *Spirulina platensis* reduced food intake, final body mass and adiposity index, and increased the total antioxidant capacity (TAC) of adipose tissue. However, no change was observed in erectile function. In the HD group, without *Spirulina* supplementation, a decrease in food intake was observed, in addition to an increase of final body mass, weight gain, adipose reserves, and adiposity index. Additionally, reduction in the number and increase in the latency of penile erection and adipose malondialdehyde levels, as well as a reduction in TCA was noted. Furthermore, cavernous contractility was increased, and the relaxing response was decreased. Interestingly, these deleterious effects were prevented by the algae at doses of 25, 50 and/or 100 mg/kg. Therefore, the supplementation with *S. platensis* prevents damages associated to a hypercaloric diet consumption and emerges as an adjuvant the prevention of ED.

## 1. Introduction

Erectile dysfunction is defined as the constant inability to achieve and/or maintain a penile erection sufficient for sexual satisfaction, and emerges among the health problems most feared by men [1,2]. Recently, the impact of obesity on male fertility has raised the population’s interest in understanding the processes underlying this problem and thus minimizing harmful changes in sexual function, such as ED [3].

From the therapeutic point of view, it is possible to reverse ED through lifestyle, use of phosphodiesterase type 5 enzyme inhibitor drugs, testosterone replacement therapy in case of hypogonadism, surgical procedures, use of vacuum constriction devices that promote erection, among others [4]. However, refractoriness to conventional treatments coupled with preference for the use of natural resources by population, due to the rescue and appreciation of popular knowledge, contributes to the search for new therapeutic alternatives for ED [5,6,7]. Recently, we established a model for the study of promising bioactive substances to treat ED in rats that is associated with hypercaloric diet consumption [8].

In this way, Spirulina platensis, a blue-green alga belonging to the cyanobacteria family that is found in both salty and sweet waters that is currently used as a food supplement for human consumption and aquaculture, stands out [9,10,11].

*S. platensis* has been highlighted due to its nutritional and medicinal potential, mainly as a consequence of its proven biological activities as a hypolipidemic in patients with hyperlipidemic nephrotic syndrome [12] or non-alcoholic fatty liver disease [13], and in rats fed with a hypercholesterolemic diet [14]; it has an anti-inflammatory effect in rats [15,16]; inhibits appetite and reduces body mass in obese patients [17,18]; acts as an antioxidant in humans and rats [19,20,21], and causes vasorelaxation in rat aorta [22]. Thus, food supplementation with *S. platensis* emerges as a potential source for the treatment of organic dysfunctions, including those affecting the smooth muscle.

Therefore, the purpose of this study was to evaluate the effect of a food supplementation with this alga on body adiposity, oxidative stress, and erectile function of Wistar rats fed with differentiated diets, a standard and a hypercaloric diet, to verify whether *r* improves penile function or prevents damage to erectile function due to increased caloric intake.

## 2. Results

### 2.1. Food Intake Evaluation

In rats fed the standard diet, the estimated mean weekly food intake of SD + SP25 (172.5 ± 2.3 g), SD + SP100 (162.5 ± 3.3 g) and SD + Sild groups (177.4 ± 2.0 g) was like the intake of the SD group (168.0 ± 2.7 g). However, this result was not reproducible in the SD + SP50 group (138.5 ± 2.3 g), which had a mean dietary intake lower than the SD group and the other experimental groups (*n* = 10).

In rats that consumed the hypercaloric diet, the estimated mean weekly food intake of the HD group (123.6 ± 1.9 g) was lower than the SD group (168.0 ± 2.7 g). When rats were supplemented with *S. platensis* at doses of 25 (157.9 ± 2.7 g) and 100 mg/kg (146.5 ± 2.2 g) or treated with sildenafil (148.9 ± 2.2 g), an increase in mean intake was observed in relation to the HD group, which was not observed for the HD + SP50 group (121.0 ± 2.5 g) that maintained this intake similar to the HD group and less than all other experimental groups (*n* = 10).

### 2.2. Caloric Intake Evaluation

Calories provided by standard diet consumption by the SD + SP25 (655.3 ± 8.6 kcal), SD + SP100 (617.5 ± 12.5 kcal) and SD + Sild groups (674.0 ± 7.6 kcal) were like those in the SD group (638.3 ± 10.1 kcal). However, this result was not reproducible in the SD + SP50 group (526.4 ± 8.8 kcal), which presented a caloric value lower than the SD group and the other experimental groups (*n* = 10).

Analyzing rats fed the high calorie diet, the calories provided by the consumption of this diet by the HD group (515.2 ± 7.8 kcal) were lower than the SD group (638.3 ± 10.1 kcal). When rats were supplemented with *S. platensis* at doses of 25 (652.9 ± 14.9 kcal) and 100 mg/kg (610.8 ± 9.1 kcal) or treated with sildenafil (621.1 ± 9.3 kcal), there was an increase in the caloric value provided by dietary consumption in relation to the HD group, which did not occur in the HD + SP50 group (506.1 ± 10.6 kcal), which maintained a lower value than the HD group and to all other experimental groups (*n* = 10).

### 2.3. Animal’s Weight Gain Evaluation

All experimental groups had similar initial body mass (Table 1 and Table 2, *n* = 10). However, rats consuming the standard diet of the SD + SP50 group (284.4 ± 10.1 g) had lower final body mass than the SD + SP25 (354.4 ± 14.3 g), SD + SP100 (338.6 ± 12.5 g) and SD + Sild groups (358.8 ± 8.7 g), not differing from the SD group (322.9 ± 4.0 g) (Table 1, *n* = 10).

In addition, the body mass gain of the SD + SP50 group (129.1 ± 9.7 g) did not differ from the SD group (166.3 ± 10.5 g) but was lower than the SD + SP25 (186.2 ± 10.0 g), SD + SP100 (176.8 ± 10.3 g) and SD + Sild groups (183.1 ± 9.2 g) (Table 1, *n* = 10).

The final body mass of the HD group (367.1 ± 12.8 g) was superior to the SD group (322.9 ± 4.0 g) and the groups fed the hypercaloric diet and supplemented with *S. platensis* at the doses of 25 (320.7 ± 11.4 g), 50 (310.2 ± 21.7 g) and 100 mg/kg (308.6 ± 9.2 g), but did not differ from the sildenafil treated group (344.4 ± 9.5 g) (Table 2, *n* = 10).

Similarly, the body mass gain of the HD group (215.0 ± 11.1 g) did not differ from the HD + Sild group (175.5 ± 6.2 g), but was greater than in the SD (166, 3 + 10.5 g), HD + SP25 (148.6 ± 14.5 g), HD + SP50 (145.3 ± 13.7 g) and HD + SP100 groups (143.1 ± 9.6 g) (Table 2, *n* = 10).

### 2.4. Dietary Efficacy, Feed Conversion and Weight Gain for Caloric Intake Coefficients

No difference was observed in the coefficients of dietary efficacy, feed conversion and weight gain for caloric intake of the rats consuming the standard diet (*n* = 10). However, rats fed the hypercaloric diet (0.22 ± 0.01) presented higher coefficient of dietary efficacy than the SD group (0.12 ± 0.01), and the rats that consumed the hypercaloric diet and were supplemented with *S. platensis* at doses of 25 (0.11 ± 0.01), 50 (0.14 ± 0.02) or 100 mg/kg (0.13 ± 0.01) or treated with sildenafil (0.15 ± 0.01) (*n* = 10).

In addition, the HD group (4.8 ± 0.3) presented a lower coefficient of feed conversion than the SD group (8.4 ± 0.6) and the groups that consumed the hypercaloric diet and received supplementation with the alga, HD + SP25 (9.6 ± 0.8), HD + SP50 (7.9 ± 1.5) and HD + SP100 (7.8 ± 0.7), not differing from the sildenafil treated group (6.8 ± 0.4) (*n* = 10).

The coefficient of weight gain for caloric intake of the HD (0.05 ± 0.01 g/kcal) and HD + Sild groups (0.06 ± 0.03 g/kcal) was higher than the SD group (0.03 ± 0.01 g/kcal). This ratio was reduced in rats fed the hypercaloric diet and supplemented with *S. platensis* at doses of 25 (0.03 ± 0.01 g/kcal), 50 (0.03 ± 0.01 g/kcal) and 100 mg/kg (0.03 ± 0.01 g/kcal) when compared to the HD group (*n* = 10).

### 2.5. Experimental Assessment of the Obesity Induction

#### 2.5.1. Murinometrics Parameters

In rats fed the standard diet, the nasoanal length of the SD + SP50 group (22.9 ± 0.4 cm) was lower than the SD + SP25 (24.4 ± 0.3 cm) and SD + Sild groups (24.5 ± 0.2 cm), not differing from the SD (23.2 ± 0.4 cm) and SD + SP100 groups (23.2 ± 0.4 cm) (*n* = 10). In rats fed the hypercaloric diet, the HD group (24.8 ± 0.3 cm) presented a nasoanal length greater than the SD group (23.2 ± 0.4 cm) and smaller than the groups fed the hypercaloric diet and supplemented with *S. platensis* at doses of 25 (23.4 ± 0.3 cm), 50 (23.2 ± 0.4 cm) and 100 mg/kg (23.2 ± 0.2 cm), not differing from the HD + Sild group (24.5 ± 0.2 cm) (*n* = 10).

In this study, no difference was observed in the Lee index of the experimental groups (*n* = 10). Among the rats fed the standard diet, BMI varied only in the SD + SP50 group (0.53 ± 0.01 g/cm^2^), where it was lower than the SD + SP100 (0.61 ± 0.02 g/cm^2^) and SD + Sild groups (0.60 ± 0.01 g/cm^2^), not differing from SD (0.58 ± 0.01 g/cm^2^) and SD + SP25 groups (0.57 ± 0.01 g/cm^2^). Despite this, the BMI of the groups that consumed the hypercaloric diet did not change (*n* = 10).

#### 2.5.2. Mass of White Adipose Tissue (WAT)

In rats fed the standard diet, the mass of epididymal adipose tissue of the SD group (1.0 ± 0.1 g/100 g) was not altered by food supplementation with *S. platensis* at doses of 25 (1.3 ± 0.1 g/100 g), 50 (1.0 ± 0.1 g/100 g) and 100 mg/kg (1.1 ± 0.05 g/100 g) or by treatment with sildenafil (1.2 ± 0.1 g/100 g). Similarly, the SD group (2.1 ± 0.2 g/100 g) presented retroperitoneal adipose tissue mass like SD + SP25 (2.1 ± 0.1 g/100 g), SD + SP50 (1.6 ± 0.2 g/100 g), SD + SP100 (1.6 ± 0.1 g/100 g) and SD + Sild groups (1.9 ± 0.1 g/100 g). However, the mass of inguinal adipose tissue was reduced in the SD + SP100 group (1.0 ± 0.05 g/100 g) when compared to SD (1.6 ± 0.1 g/100 g), SD + SP25 (1.4 ± 0.1 g/100 g) and SD + Sild groups (1.4 ± 0.1 g/100 g), not differing from SD + SP50 group (1.2 ± 0.1 g/100 g) (Figure 1A, *n* = 10).

In the rats that consumed the hypercaloric diet, the mass of the epididymal adipose tissue of the HD (2.0 ± 0.1 g/100 g) and HD + Sild groups (2.0 ± 0.1 g/100 g) was higher than the SD group (1.0 ± 0.1 g/100 g), not differing from the HD + SP25 (1.6 ± 0.1 g/100 g), HD + SP50 (1.6 ± 0.2 g/100 g) and HD + SP100 groups (1.5 ± 0.1 g/100 g). In addition, the mass of the retroperitoneal adipose tissue of the HD group (3.5 ± 0.3 g/100 g) was higher than the SD group (2.1 ± 0.2 g/100 g). Food supplementation with *S. platensis* at doses of 25 (2.6 ± 0.3 g/100 g), 50 (2.3 ± 0.3 g/100 g) and 100 mg/kg (2.5 ± 0.3 g/100 g) or treatment with sildenafil (2.6 ± 0.2 g/100 g) did not alter the mass of this adipose reserve. Additionally, the inguinal adipose tissue mass was increased in the HD group (2.4 ± 0.2 g/100 g) when compared to the SD group (1.6 ± 0.1 g/100 g), showing no difference for HD + Sild (2.0 ± 0.2 g/100 g). However, supplementation with the algae at doses of 25 (1.7 ± 0.2 g/100 g), 50 (1.6 ± 0.1 g/100 g) and 100 mg/kg (1.5 ± 0.1 g/100 g) prevented the increase in this adipose tissue deposit when compared to the HD group (Figure 1B, *n* = 10).

#### 2.5.3. Body Adiposity Index

In rats fed the standard diet, the SD group (1.5 ± 0.1) presented with similar body adiposity index to the SD + SP25 (1.4 ± 0.1), SD + SP50 (1.5 ± 0.1) and SD + Sild groups (1.3 ± 0.06). However, the SD + SP100 group (1.0 ± 0.1) presented a reduction in this index when compared to the SD, SD + SP25 and SD + SP50 groups (*n* = 10).

The rats that consumed the hypercaloric diet (2.2 ± 0.1) had a body adiposity index higher than the SD group (1.5 ± 0.1), and the rats supplemented with *S. platensis* at doses of 25 (1.6 ± 0.1), 50 (1.6 ± 0.1) and 100 mg/kg (1.6 ± 0.2), differing from rats treated with sildenafil (1.9 ± 0.1). The groups supplemented with the alga, on the other hand, presented no difference in relation to the SD group (*n* = 10).

#### 2.5.4. Biochemical Analysis

In this study, no differences were observed in the biochemical parameters of rats consuming the standard or hypercaloric diets (Table 3, *n* = 10).

### 2.6. Penile Erection Induction

No difference was observed in the number of penile erections and the erection latency among the groups fed the standard diet (Figure 2A, *n* = 6). The number of penile erections in the SD group (2.5 ± 0.2) was higher than the HD group (0.5 ± 0.2). When rats consumed the hypercaloric diet and were supplemented with *S. platensis* at doses of 25 (1.2 ± 0.3) and 50 mg/kg (1.5 ± 0.2), no difference was observed for the HD group. In addition, the HD + SP100 (1.7 ± 0.3) and HD + Sild groups (1.7 ± 0.2) increased the number of penile erections when compared to the HD group, not differing from the SD group (Figure 2B, *n* = 6).

The latency to obtain a penile erection in the HD group (23.5 ± 3.1 min) was superior to the SD group (12.8 ± 1.9 min). In rats fed the hypercaloric diet, compared to the HD group, this parameter was not altered by supplementation with 25 mg/kg of *S. platensis* (15.8 ± 2.9 min), but was reduced by supplementation with 50 (9.3 ± 2.6 min) and 100 mg/kg (9.8 ± 1.1 min) or by treatment with sildenafil (7.0 ± 0.4 min), not differing from the SD group (Figure 2B, *n* = 6).

#### Correlation between the Parameters of Experimental Obesity and the Erectile Function of Rats

The negative correlation of the number of penile erections with the final body mass (*r* = −0.22, *p* = 0.20) and the relative mass of the epididymal (*r* = −0.32, *p* = 0.06), retroperitoneal (*r* = −0.28, *p* = 0.09) and inguinal adipose tissues (*r* = −0.21, *p* = 0.22) was not significant. However, a moderate negative correlation was observed between the number of penile erections with the adiposity index (*r* = −0.43, *p* = 0.01) and body mass gain (*r* = −0.48; *p* = 0, 01) (Figure 3, *n* = 6).

The positive correlation of latency for penile erection with final body mass (r = 0.32, *p* = 0.06) and the relative mass of epididymal (r = 0.03, *p* = 0.86), retroperitoneal (r = 0.33, *p* = 0.06) and inguinal adipose tissues (r = 0.16, *p* = 0.36) was not significant. However, a weak positive correlation of latency to obtain penile erection with adiposity index (r = 0.37, *p* = 0.03) and body mass gain (r = 0.30, *p* = 0.02) (Figure 4, *n* = 6).

### 2.7. Corpus Cavernosum Reactivity

#### 2.7.1. Contractile Reactivity Measurement

In the groups fed the standard diet, supplementation with *S. platensis* at doses of 25 (E_max_ = 99.3 ± 2.0%; pD_2_ = 5.4 ± 0.04) and 100 mg/kg (E_max_ = 86.6 ± 4.5%; pD_2_ = 5.8 ± 0.2) or treatment with sildenafil (E_max_ = 101.9 ± 13.1%; pD_2_ = 5.8 ± 0.04) did not change the contractile reactivity of the corpus cavernosum to Phe, since no difference was observed when compared to the SD group (E_max_ = 100%; pD_2_ = 5.5 ± 0.06). However, supplementation of rats with 50 mg/kg of *S. platensis* reduced the pD_2_ of Phe (pD_2_ = 4.8 ± 0.2) and increased the contractile efficacy of this agonist (E_max_ = 156.6 ± 11.6%) (Figure 5A, *n* = 5).

In rats fed the hypercaloric diet, the contractile potency of Phe was reduced in the HD group (pD_2_ = 4.8 ± 0.06) when compared to the SD group (pD_2_ = 5.5 ± 0.06). Additionally, the E_max_ of this agonist was increased in the HD group compared to the SD group (E_max_ = 147.5 ± 11.2 and 100% respectively). However, supplementation with doses of 25 (E_max_ = 84.5 ± 3.0%; pD_2_ = 5.3 ± 0.07), 50 (E_max_ = 72.7 ± 5.6%; pD_2_ = 5, 6 ± 0.1) and 100 mg/kg of *S. platensis* (E_max_ = 79.6 ± 6.1%; pD_2_ = 5.3 ± 0.1), as well as treatment with sildenafil (E_max_ = 104.2 ± 9.7%; pD_2_ = 5.5 ± 0.1) reduced the contractile efficacy and increased the potency of Phe in the rat corpus cavernosum when compared to the HD group, not differing from the SD group (Figure 5B, *n* = 5).

#### 2.7.2. Relaxing Reactivity Measurement

Analyzing the relaxation of the corpus cavernous promoted by ACh in the groups fed the standard diet, supplementation with *S. platensis* at doses of 25 (E_max_ = 69.6 ± 2.5%; pD_2_ = 7.7 ± 0.1) and 100 mg/kg (E_max_ = 64.1 ± 2.6%; pD_2_ = 7.4 ± 0.2) did not change the relaxing reactivity of the rat corpus cavernosum to ACh, since no difference was observed when compared to the SD group (E_max_ = 72.7 ± 3.5%; pD_2_ = 7.6 ± 0.1). However, supplementation of rats with 50 mg/kg of *S. platensis* or treatment with sildenafil increased the relaxing efficacy of this agonist (E_max_ = 94.4 ± 5.3 and 92.7 ± 3.1%, respectively), without a change in ACh potency (pD_2_ = 7.7 ± 0.2 and 7.4 ± 0.3, respectively) when compared to the SD, SD + SP25 and SD + SP100 groups (Figure 6A, *n* = 5).

Furthermore, the relaxing efficacy of ACh was decreased in the HD group (E_max_ = 50.7 ± 2.3%) when compared to the SD group (E_max_ = 72.7 ± 3.5%); however, the agonist potency was similar in both HD and SD groups (pD_2_ = 7.0 ± 0.1 and 7.6 ± 0.1, respectively) (Figure 6B, *n* = 5).

In the groups fed the hypercaloric diet, supplementation with *S. platensis* at a dose of 25 mg/kg (E_max_ = 53.6 ± 2.8%; pD_2_ = 7.0 ± 0.1) did not alter the relaxation promoted by ACh in the rat corpus cavernosum when compared to the HD group (E_max_ = 50.7 ± 2.3%; pD_2_ = 7.0 ± 0.1). However, supplementation with 50 mg/kg (E_max_ = 70.3 ± 5.7%; pD_2_ = 9.0 ± 0.1) or treatment with sildenafil (E_max_ = 97.6 ± 2.4%; pD_2_ = 8.6 ± 0.1) increased the potency and relaxing efficacy of ACh when compared to the HD group. In addition, the relaxing potency of this agonist was increased in the HD + SP100 group (pD_2_ = 7.9 ± 0.3); however, with no change in the contractile efficacy of ACh (E_max_ = 44.4 ± 4.3%), comparing to the HD group. Additionally, the best relaxing efficacy promoted by ACh was observed in the HD + Sild group; despite this, the agonist potency was higher in both HD + SP50 and HD + Sild groups when compared to the other experimental groups (Figure 6B, *n* = 5).

## 3. Discussion

In the present study, using a model of erectile dysfunction associated with the consumption of a hypercaloric diet in Wistar rats, it was observed that the hypercaloric diet intake resulted in the increase of body mass and parameters related to body adiposity, as well as being correlated with the reduction of erectile function. Interestingly, these effects were prevented in rats fed the hypercaloric diet by food supplementation with *S. platensis*.

*S. platensis*, which is a bluish-green algae with high nutritional value, has aroused the interest of researchers as a potential source for the treatment of different diseases [23,24,25]. Recently, it has been shown that dietary supplementation with this algae has promoted beneficial effects in models of vascular smooth muscle by reducing contraction and increasing relaxation of the aorta and ileum of Wistar rats [22,26].

Thus, we decided to evaluate the impact that the consumption of the high calorie diet would promote on the erectile function of rats and to verify if the supplementation with *S. platensis* would prevent the development of the physiological dysfunctions triggered by the dietary alteration. 

In this sense, comparing the consumption of the different diets, the HD group presented estimated food intake and caloric value provided by the consumption of the diet inferior to the SD group. These data agree with different studies that used the same high calorie diet [27,28]. Due to the high energy efficiency of hyperlipidic and hypercaloric diets, it is common to have a reduction in the food intake of the animals in relation to those who consumed a standard diet [29,30]. Furthermore, in diets with a high percentage of lipids, such as that used in this study (16 vs. 4% of the standard diet), this decrease of food consumption is also attributed to the increased secretion of cholecystokinin (CCK), a hormone that induces satiety [31,32], a fact that may justify the differences observed in the rats’ food consumption in the groups HD and SD.

Interestingly, among the groups that simultaneously received the hypercaloric diet and supplementation with *S. platensis*, HD + SP25 and HD+ SP100 showed an increase in the estimated food consumption and in the caloric value provided by the diet consumption, while the HD + SP50 group did not change these parameters when compared to the HD group. In this context, it should be noted that the effect of the algae on food intake in the groups that consumed this diet can be associated, in some cases, with the negative modulation of hormones that stimulate satiety, but further studies are needed to prove these data.

Rats that consumed the hypercaloric diet and were treated with sildenafil also had an increase in estimated food intake and in the caloric amount provided by the diet. In this case, these were not correlated with the erectogenic mechanism of action of sildenafil, but rather highlighted as an adverse reaction [33].

In addition, all groups had a similar body mass range at the beginning of the experimental protocol (Table 1 and Table 2). However, when comparing the groups that consumed the standard and hypercaloric diets, the HD group had higher final body mass and mass gain than the SD group (Table 2), corroborating different studies that used the same diet used in this study with rats [27,34,35].

In rats that consumed the hypercaloric diet and received supplementation with the algae simultaneously, there was a reduction in final mass and in body mass gain in relation to the HD group (Table 2) that did not differ from the SD group. The increase in food intake and the reduction in body mass gain in rats seems contradictory; however, the protein composition (65 70%) of *S. platensis* can help increase thermogenesis. Different studies suggest that the increase in protein intake triggers the loss of body mass and hinders its recovery due to increased protein synthesis and consumption of ATP for the synthesis of peptide bonds [36,37].

In addition, treatment with sildenafil in rats fed the hypercaloric diet did not alter the final mass or gain in body mass of the rats, even increasing the food intake of these animals (Table 2). The ability of sildenafil to raise cGMP levels favors PKG activation and, consequently, stimulates the transdifferentiation of white adipocytes in the intermediate (brown-white) or beige phenotype, which is more metabolically active [38].

In this view, there are different factors that influence the body mass gain of animals, such as the ability to eat food, to transform the diet provided by selecting the material ingested, or to make better use of the food eaten or the genetic potential for accumulation of body mass (limiting factor). Given this information, the evaluation of coefficients such as the coefficients of dietary efficacy, feed conversion and weight gain for caloric intake stands out to determine the animal’s efficiency in converting the feed consumed into body mass [39,40].

Given these premises, supplementation with *S. platensis* or treatment with sildenafil did not alter the coefficients of dietary efficacy, feed conversion and weight gain for caloric intake from rats that consumed the standard diet (Table 3). However, the HD group showed an increase in coefficients of dietary efficacy and weight gain for caloric intake and a reduction in the coefficient of feed conversion when compared to the SD group, demonstrating a better performance in converting the feed consumed into body mass, as well as justifying the increase in body mass of rats, even with a lower caloric intake. In addition, simultaneous supplementation with *S. platensis* in the three doses reduced the coefficients of dietary efficacy and weight gain for caloric intake and increased the coefficient of feed conversion of rats that consumed the hypercaloric diet when compared to the HD group, not differing from the SD group, suggesting less conversion efficiency of the diet consumed in body mass, which can support the decrease in final mass and gain in body mass in the HD + SP25, HD + SP50 and HD + SP100 groups.

Furthermore, treatment with sildenafil, in the HD + Sild group, reduced the coefficient of dietary efficacy compared to the HD group, not differing from the SD group, demonstrating a lower utilization of the feed consumed which, compensated by the increased food consumption of these rats, may justify the non-alteration of body mass gain in this group (Table 2).

In the analysis of murinometric parameters, no differences were observed between the Lee index of the experimental groups, like that observed in different studies with animal obesity induced by excessive intake of sucrose or lipids [41]. Among the groups analyzed in this study, only the SD + SP100 and SD + Sild groups had a higher BMI than the SD + SP50 group, justified by the increase in final body mass and mass gain in these experimental groups. The animals’ BMI ranged from 0.53 to 0.61 g/cm^2^, within the range considered normal for rats aged 30 to 150 days (0.38–0.68 g/cm^2^). In this scenario, the change in the nasoanal length of the rats, since the animals are in the growth phase, can be correlated to the non-variation of these indices for most experimental groups. In addition, the murinometric changes of rats associated with the development of obesity are observed in an average period of 12 weeks of consumption of high-fat or high-calorie diets [42].

Another aspect that stands out in the determination of physiological dysfunctions resulting from the ingestion of hyperlipidic or hypercaloric diets is adiposity, verified through the direct quantification of visceral and subcutaneous adipose reserves [42]. In rats fed the standard diet, a reduction in inguinal adipose tissue and inguinal adipocyte diameter was observed in the group supplemented with a dose of 100 mg/kg of *S. platensis* (Figure 1), demonstrating prevention in subcutaneous adipose deposition, which may be related to reduced triglyceride uptake.

Furthermore, in rats fed the hypercaloric diet, all adipose deposits and adipocyte diameters in the HD group were superior to the SD group (Figure 1), corroborating the previous results that demonstrated an increase in the final body mass of the HD group in relation to the SD group. In this scenario, the increase in the proportion of lipids present in the hypercaloric diet (16% vs. 4% of the standard diet) increases their feeding efficiency due to the low thermal effect of the lipids to be metabolized. Thus, there is a greater deposition of fatty acids that are not used for energy production, in the form of triglycerides, in adipocytes [43].

In rats fed the hypercaloric diet and supplemented with the algae, only a reduction in the inguinal adipose reserve and inguinal adipocyte diameter was observed when compared to the HD group, not differing from the SD group (Figure 1), suggesting a prevention of subcutaneous adipose deposition, which is less harmful to the body [44].

In the assessment of the adiposity index, the SD + SP100 group showed a reduction in the adiposity index when compared to the other groups fed the standard diet, which can be justified by the reduction of the inguinal adipose deposit. However, after consuming different diets, the HD group presented this parameter higher than did the SD group due to the increase in the deposition of lipids in all adipose reserves. Additionally, with the food supplementation with *S. platensis* in the three doses, there was a reduction of this index in relation to the HD group, not differing from the SD group, which can be attributed to the decrease in the mass of the inguinal adipose tissue (subcutaneous).

In addition, it was verified that the biochemical profile of Wistar rats was not altered by the consumption of the hypercaloric diet or by food suplementation with *S. platensis* (Table 3). Similar results were obtained in other studies in which the rats were fed a hypercaloric diet [27,45,46,47,48].

In summary, food supplementation with *S. platensis* prevented physiological dysfunctions in the body adiposity of rats that consumed the hypercaloric diet. Recently, Souza et al. [8,49] showed a decrease in the cavernous and intestinal contractile reactivity in rats fed for eight weeks with a hypercaloric diet. In view of these premises, it was hypothesized that the consumption of the hypercaloric diet would impair the erectile function of Wistar rats, and that food supplementation with *S. platensis* would prevent this harmful effect.

In this study, the erectile function of groups fed the standard diet was similar in the absence or presence of supplementation with the algae or treatment with sildenafil (Figure 2), showing that *S. platensis* and sildenafil do not suppress the erectogenic mechanism. On the other hand, the HD group showed a reduction in the number of penile erections and an increase in the latency to start the erection when compared to the SD group (Figure 2), suggesting the development of ED in these rats, because some of the main symptoms of this disease are the difficulty to achieve and/or maintain an erection, as well as the increase in the time needed to start the penile erection [50].

In rats supplemented with *S. platensis*, there was an increase in the number of penile erections at a dose of 100 mg/kg and a reduction in the time to start penile erection at doses of 50 and 100 mg/kg. In addition, chronic treatment with sildenafil, a standard drug for ED therapy, in the HD + Sild group, also increased the number of erections and reduced latency for penile erection (Figure 2). These results are attributed to the inhibition of PDE5 enzymes, leading to an increase in intracellular levels of cGMP in the cavernous bodies of the penis, which favors cavernous relaxation and penile erection [51].

In addition, Pearson’s correlation [52] was used to measure the degree of relationship between the changes in the erectile function of the rats with the parameters affected by the consumption of the hypercaloric diet. In this analysis, the number of penile erections showed a negative and moderate correlation with the adiposity index and the body mass gain of the rats (Figure 3), demonstrating that the decrease in penile erections can be associated with the increase in total body adiposity, but not to a deposit of specific adipose tissue, as well as to excessive gain of body mass. Also, the time required for the initiation of penile erection showed a positive and weak correlation with the adiposity index and the body mass gain of the rats (Figure 4), that is, the latency increases with the increase in total body adiposity and the gain in the body mass of rats.

In view of these results, the negative modulation of obesity on the penile erection process in rats is demonstrated, and it is proposed that food supplementation with *S. platensis*, by reducing the parameters of experimental obesity, prevents damage to the erectile function of these rats. However, one cannot rule out the possibility that the algae also modulate parallel targets that result in improved erectile function in these rats, such as the reactivity of cavernous smooth muscle cells.

Once an alteration in the erectile function of rats was verified through in vivo studies, because of the consumption of differentiated diets, it was hypothesized that the consumption of the hypercaloric diet could increase the contractile reactivity and reduce the relaxing response of the rat corpus cavernosum.

In this context, when comparing the Phe curve between the HD and SD groups, it was observed that the relative contractile potency of the agonist was attenuated in the HD group in relation to the SD group. However, the maximum contractile response of Phe was increased in the HD group (Figure 5). These results may be associated with the increase in mechanisms that favor cavernous contractility, such as increased expression of Ca_V_ and/or positive modulation of the RhoA/ROCK pathway, and, consequently, penile flaccidity. Allied to these results, there are reports of increased contractile efficacy of Phe in the corpus cavernosum associated with the expression of ROCK2 in obese mice [53], diabetic rats [54] and elderly mice [55].

Furthermore, dietary supplementation with all doses of *S. platensis* or treatment with sildenafil potentiated the contractile effect of Phe; however, this resulted in a reduction in the effectiveness of this contractile agent (Figure 5). Therefore, it is suggested that the supplements used in this study can reduce steps of cavernous contraction pathways, as well as activate steps of signaling pathways that culminate in the relaxation of the rat corpus cavernosum and, in this case, prevent the changes triggered by the consumption of hypercaloric diet on cavernous contractile reactivity.

Additionally, the endothelial dysfunction resulting from the consumption of a hypercaloric diet plays a central role in the regulation of rat corpus cavernosum reactivity and can directly influence the sexual function of these rats. In this context, to verify a possible modulation of the food intake of the hypercaloric diet and of the supplementation with *S. platensis* on the endothelium-dependent relaxing responsiveness, cumulative relaxation curves with ACh were performed.

The data obtained with this methodology demonstrate that among the rats that consumed the standard diet, only those that received supplementation with *S. platensis* at a dose of 50 mg/kg or were treated with sildenafil showed an increase in the relaxing efficacy promoted by ACh (Figure 6), indicating that through the positive modulation of endothelium-derived relaxing factors, the algae supplementation can increase rat corpus cavernosum relaxation.

Furthermore, the treatment of rats that consumed the standard diet with sildenafil increased the relaxing efficacy promoted by ACh, a fact expected due to the activation of the NO/cGMP/PKG pathway [56].

When comparing the ACh relaxation curve obtained in the HD and SD groups, it was shown that the relative relaxing efficacy of the agonist was reduced in the HD group (Figure 6). Different studies demonstrate the interconnection between endothelial dysfunction and the reduction of endothelium-dependent relaxation in the corpora cavernosa of obese [49], diabetic [50] or elderly mice [51].

Food supplementation with 50 and 100 mg/kg of *S. platensis* and treatment with sildenafil potentiated the relaxing effect of ACh in the corpora cavernosa of rats fed a hypercaloric diet (Figure 6). In view of this, it is suggested that these supplements can modulate steps of the NO and/or prostanoid signaling pathways, a fact that favors the relaxation of the rat corpus cavernosum and, in this case, prevents the changes triggered by the ingestion of a hypercaloric diet on the cavernous relaxing reactivity.

As an outcome, based on the points analyzed in this study, the limitations of the study should be minimized later, since it is proposed to carry out additional investigations, at the functional and molecular level, as well as clinical trials, to verify if these data are transposable for humans, since based on the relationship of body surface area, the food supplements used in this study (25, 50 and 100 mg/kg) are within an acceptable profile for use in humans (280, 560 and 1120 mg/70 kg, respectively) and, in this way, confirm the beneficial potential of food supplementation with *S. platensis* on the development of erectile dysfunction associated with the food intake of a hypercaloric diet.

## 4. Materials and Methods

### 4.1. Test Product

*Spirulina platensis* was obtained as powder (product batch n°. 2014007) from Dongtai Spirulina Bio-Engineering Co., Ltd. (Nanjing, China). A sample was analyzed by the Quality Control of Galena Química e Farmacêutica Ltd. a Laboratory, Campinas, Brazil (product batch No. 1504023402) for the certification that the material was *S. platensis* and marketed by Dilecta Farmácia de Manipulação e Homeopatia (João Pessoa, Paraíba, Brazil) (product batch No. 2014007). This species is cultivated in marine environments and outdoors, with the purpose of being used as a food supplement (product batch No. 1504023402).

### 4.2. Animals

For protocols, male Wistar rats (*Rattus norvegicus*) of approximately 160 g (8 weeks of age) were used, obtained from the Unidade de Produção Animal Prof. Thomas George from the Instituto de Pesquisa em Fármacos e Medicamentos (IPeFarM) of Universidade Federal da Paraíba (UFPB), João Pessoa, Brazil. The rats were maintained in a 12-h light-dark cycle under controlled ventilation and temperature (21 ± 1 °C) with free access to water. The experiments were performed following the principles of guidelines for the ethical use of animals in applied etiology studies [57] and the National Council for the Control of Animal Experimentation (CONCEA) [58], andwere approved by the Ethics Committee on Animal Use of UFPB (protocol No, 0201/14).

### 4.3. Diets

The standard diet (Presence^®^) containing by weight 23% protein, 63% carbohydrate and 4% lipids with an energy density 3.8 kcal/g, and the hypercaloric diet was composed of a diet (Presence^®^) of milk chocolate, peanuts and sweet biscuits (3:2:2:1 proportion). The preparation was done by mixing all the powdered components. The diet was prepared weekly and supplied to rats in the form of pellets. The analysis of the hypercaloric diet nutritional composition was performed at the Laboratório de Nutrição Experimental from UFPB, João Pessoa, Brazil, and contained by weight 22.8% protein, 45.5% carbohydrate and 16% lipids, with an energy density 4.2 kcal/g. Each experimental group was fed for eight weeks [8].

### 4.4. Experimental Groups

The rats were randomly divided into 10 experimental groups: groups fed with a standard diet and supplemented with saline solution (NaCl 0.9%) (SD) or *S. platensis* 25 (SD + SP25), 50 (SD + SP50) and 100 mg/kg (SD + SP100) or treated with sildenafil 1.5 mg/kg (SD + Sild); groups fed with a hypercaloric diet and supplemented with saline solution (NaCl 0.9%) (HD) or *S. platensis* 25 (HD + SP25), 50 (HD + SP50) and 100 mg/kg (HD + SP100) or treated with sildenafil 1.5 mg/kg (HD + Sild). The *S. platensis* powder was dissolved in saline solution (0.9% NaCl) daily for the preparation of the doses used in the study, which were administered to rats at the end of the preparation.

### 4.5. Chemicals

The R-(-)-apomorphine, malondialdehyde (MDA), 1,1-diphenyl-2-picryl hydrazyl (DPPH) and acetylcholine (ACh) were obtained from Sigma-Aldrich (São Paulo, Brazil). Phenylephrine (Phe) was purchased from Pfizer (Kalamazoo, USA). Magnesium sulphate (MgSO_4_), potassium chloride (KCl), calcium chloride (CaCl_2_), sodium chloride (NaCl) and formaldehyde were purchased from Vetec Química Fina Ltda. (São Paulo, Brazil). Glucose (C_6_H_12_O_6_) and sodium bicarbonate (NaHCO_3_) were purchased from Dinâmica (São Paulo, Brazil). Potassium monobasic phosphate (KH_2_PO_4_), sodium hydroxide (NaOH) and hydrochloric acid (HCl) were purchased from Nuclear (São Paulo, Brazil). These substances, except the glucose, NaCl and NaHCO_3_ were diluted in distilled water to obtain each solution, which were maintained under refrigeration.

### 4.6. Food Intake and Animal’s Weight Gain Evaluation

The food intake was calculated daily as the difference between the offered and residual food. The food that was not eaten and remained in the outer area of the cage was considered as clean reject [59]. The body mass (g) of the animals was recorded weekly, and the weight gain was calculated by the difference between the final and initial body mass.

### 4.7. Dietary Efficacy, Feed Conversion and Weight Gain for Caloric Intake Coefficients

The dietary efficacy coefficient was calculated by the ratio between the body mass gain (g) and the total food intake (g) of the rats after the experimental period. The feed conversion coefficient was calculated by the ratio between total food intake (g) and body weight gain (g) after the experimental period. The coefficient of weight gain per caloric consumption was calculated by the ratio between the body mass gain (g) and the total energy consumption (kcal) of the rats after the experimental period [60].

### 4.8. Experimental Assessment of the Obesity Induction

#### 4.8.1. Murinometrics Parameters

At the end of treatments, rats were weighed, and the naso-anal length was measured to calculate the Lee index as the cubic root of the body mass divided by the naso-anal length [49] and the BMI as the ratio between body weight (g) and the square of body length (cm^2^) [61].

#### 4.8.2. Mass of White Adipose Tissue

Twenty-four hours after the last exposure to diet and/or supplementation, rats were euthanized by guillotine and the WAT was carefully dissected and weighed, particularly the adipose inguinal, retroperitoneal and epididymal tissues, since they represent the main constituents of the central adiposity.

#### 4.8.3. Body Adiposity Index

The body adiposity index was calculated using the formula: inguinal + retroperitoneal + epididymal fat × 100/final body weight [62].

#### 4.8.4. Adipose Tissue Morphometry

WAT samples were fixed in formaldehyde (10%), subjected to a standard histological procedures. Briefly, dehydration was undertaken by (1) increasing the alcohol series of 70% for 24 h, 80, 96, and 100% (third bath) for 1 h each; diaphanization (2), by bath in 100% xylene alcohol (1:1) for 1 h, followed by two baths in pure xylene for 1 h each; and embedding in paraffin; (3) by passing the WAT through two baths of liquid paraffin (heated to 50 °C) for 1 h each. Next, WAT were embedded in a new paraffin. The blocks obtained were cut to 5 μm thick and stained with Mayer’s hematoxylin/eosin [63]. Digital images of histological sections were obtained, and about 100 adipocytes per animal had its diameter measured using the Leica Qwin 3.1 software (Wetzlar, Germany).

### 4.9. Biochemical Analysis

After 12 h fasting, the rat blood was collected by cardiac puncture. Therefore, blood was placed in test tubes containing EDTA to obtain the plasma [64]. Then, samples were centrifuged at 0.02 G for 15 min and the supernatant was transferred to Eppendorfs^®^ at –80 °C. Glucose, triglyceride, total cholesterol, and high density lipoprotein (HDL-c) fractions were performed using specific commercial kits Labtest^®^ (Minas Gerais, Brazil), according to manufacturer standards, on the automatic biochemical analyzer LabMax 240 (Minas Gerais, Brazil). Low-density lipoprotein (LDL-c) fractions were estimated by the Friedewald, Levy and Fredrickson (1972) [65] equation: [LDL-c = (total cholesterol − HDL-c) − (triglycerides/5)].

### 4.10. Penile Erection Induction

Rats were individually placed in a box for 30 min, for acclimatization. Next, animals received a dorsal subcutaneous injection of apomorphine (80 mg/kg) prepared in saline solution (NaCl 0.9%). Then, rats were filmed during 30 min and through the images the latency time for achieve the first erection and the number of erections obtained for each animal were measured. Erections were events where it was possible to observe lordosis, in which the animal rests on its hind legs, leaning the body forward, holding the penis and licking the organ and ends with penile erection [66].

### 4.11. Corpus Cavernosum Reactivity

Animals were euthanized by guillotine and the corpus cavernosum was immediately removed, cleaned of fat and connective tissue, immersed in physiological solution at room temperature and bubbled with a carbogen mixture. To register isometric contractions, corpus cavernosum segments (1 cm) were suspended in steel rods in organ baths (6 mL), connected to a force transducer (TIM 05), attached to an amplifier (AECAD04F) and connected to an A/D converter into a PC running AQCAD^®^ software (São Paulo, Brazil). The system contained a thermostatic pump model BT 60 that controlled the organ baths temperature.

The physiological solution used was Krebs solution, whose pH was adjusted to 7.4, and the composition (in mM) was: NaCl (118.4), KCl (4.7), CaCl_2_ (2.5), MgSO_4_ (1.2), NaHCO_3_ (25.0), KH_2_PO_4_ (1.17), d-glucose (5.6). The corpus cavernosum was stabilized for 1 h under a resting tension of 0.5 g at 37 °C and bubbled with a carbogen mixture [67].

#### 4.11.1. Contractile Reactivity Measurement

The corpus cavernosum was assembled as described in Section 4.11. After the stabilization period, cumulative concentration-response curves were obtained to Phe (10^−8^ × 10^−3^ M). The contractile reactivity was evaluated based on the values of the negative logarithm of the molar concentration of a substance that produced 50% of its maximal effect (pD_2_) and the maximum effect (E_max_) of the contractile agent, calculated from the concentration-response curves obtained.

#### 4.11.2. Relaxing Reactivity Measurement

The corpus cavernosum was assembled as described in Section 4.11. After the stabilization period, a contraction was induced with Phe 10^−5^ M. Then, a cumulative concentration-response curve was performed to ACh (10^−12^–10^−3^ M). The relaxing reactivity was evaluated based on the values of pD_2_ and E_max_, calculated from the concentration-response curves obtained.

### 4.12. Statistical Analysis

Data were expressed as the mean and standard error of the mean (S.E.M.), and statistically analyzed by one-way analysis of variance (ANOVA) followed by Tukey’s post-test. For correlation between the variables, the Pearson’s correlation coefficient (r) was used. Values were significantly different when *p* < 0.05. All data were analyzed by GraphPad Prism^®^ version 5.01 (GraphPad Software Inc., San Diego, CA, USA).

## 5. Conclusions

In this study, the negative modulation of obesity over the penile erection process of rats was demonstrated and food supplementation with *S. platensis* was applied by reducing the levels of previous experimental obesity or damage to the erectile function of these rats. However, it is not possible to rule out the possibility of algae also modulating the parallels that result in improvements in the erectile function of these rats, such as the reactivity of cavernous smooth muscle cells, as demonstrated in the experiments.

## Figures and Tables

**Figure 1 marinedrugs-20-00467-f001:**
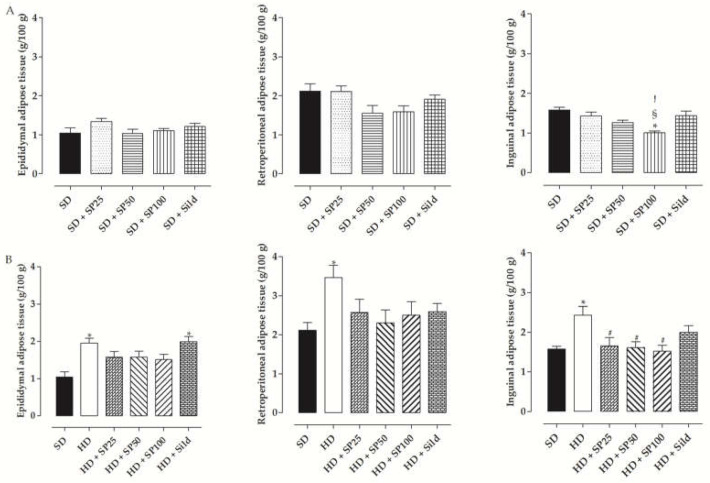
Mass of the epididymal, retroperitoneal and inguinal adipose tissues of rats from the SD, SD + SP25, SD + SP50, SD + SP100, SD + Sild (**A**), HD, HD + SP25, HD + SP50, HD + SP100 and HD + Sild groups (**B**). The columns and vertical bars represent the mean and S.E.M. respectively (*n* = 10). One way ANOVA followed by Tukey’s post-test. * *p* < 0.05 (SD vs. SD + SP100, HD and HD + Sild); § *p* < 0.05 (SD + SP25 vs. SD + SP100); ! *p* < 0.05 (SD + Sild vs. SD + SP100) and # *p* < 0.05 (HD vs. HD + SP25, HD + SP50 and HD + SP100).

**Figure 2 marinedrugs-20-00467-f002:**
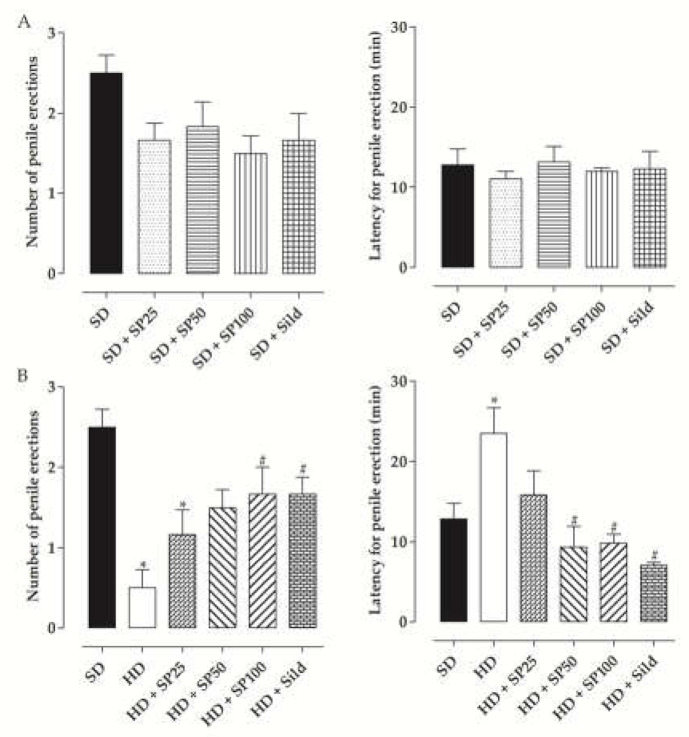
Number of penile erections and latency for penile erection of rats from the SD, SD + SP25, SD + SP50, SD + SP100, SD + Sild (**A**), HD, HD + SP25, HD + SP50, HD + SP100 and HD + Sild groups (**B**). The columns and vertical bars represent the mean and S.E.M. respectively (*n* = 6). One-way ANOVA followed by Tukey’s post-test. * *p* < 0.05 (SD vs. HD and HD + SP25) and # *p* < 0.05 (HD vs. HD + SP50, HD + SP100 and HD + Sild).

**Figure 3 marinedrugs-20-00467-f003:**
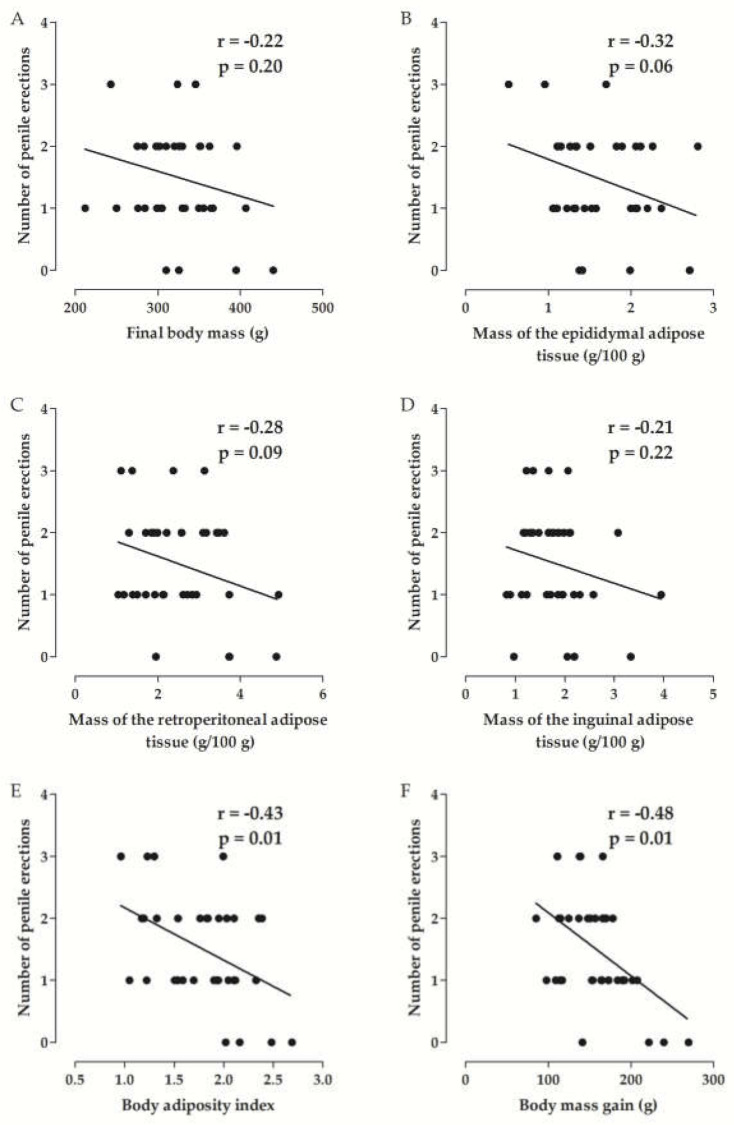
Correlation between the number of penile erections and the final body mass (**A**), the mass of epididimal (**B**), retroperitoneal (**C**) and inguinal adipose tissue (**D**), adiposity index (**E**) and body mass gain (**F**) of rats from the SD, HD, HD + SP25, HD + SP50, HD + SP100 and HD + Sild rats.

**Figure 4 marinedrugs-20-00467-f004:**
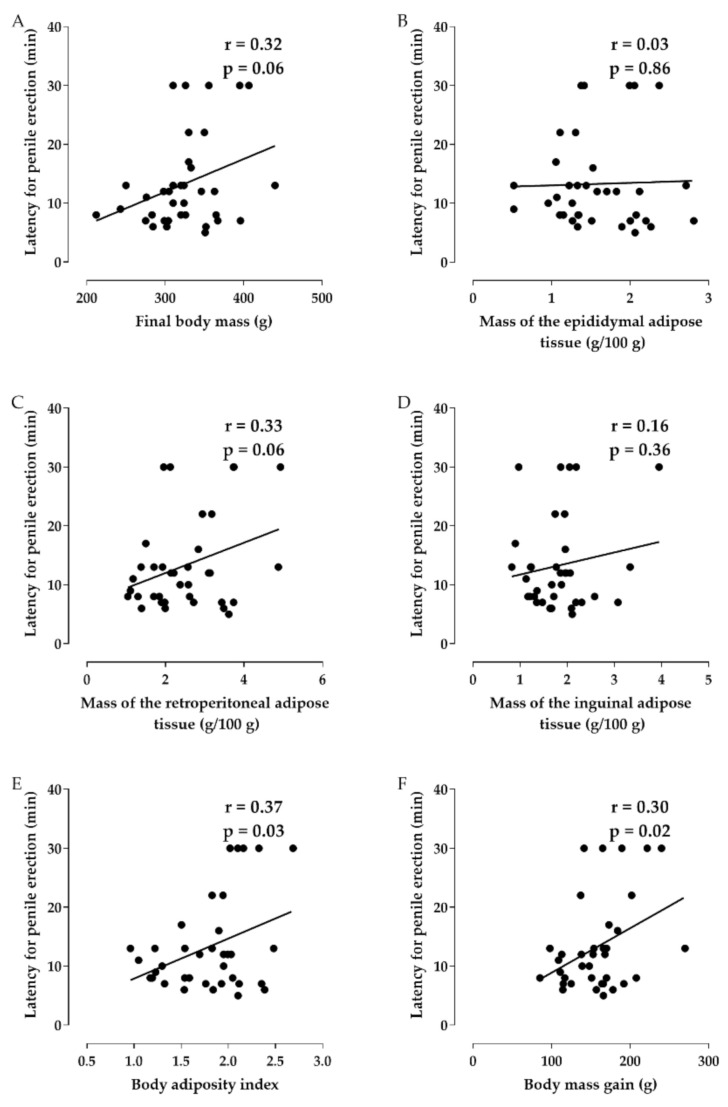
Correlation between the latency for penile erection and the final body mass (**A**), the mass of epididimal (**B**), retroperitoneal (**C**) and inguinal adipose tissue (**D**), adiposity index (**E**) and body mass gain (**F**) of rats from the SD, HD, HD + SP25, HD + SP50, HD + SP100 and HD + Sild rats.

**Figure 5 marinedrugs-20-00467-f005:**
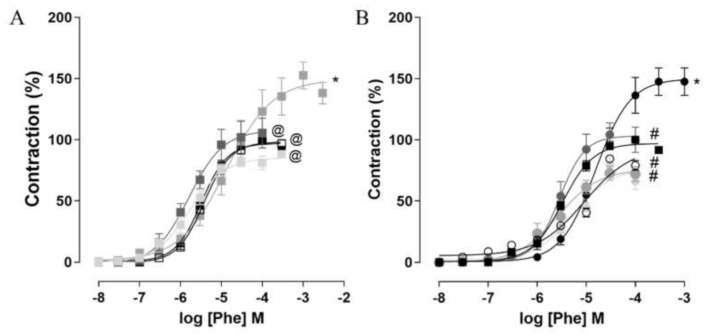
Cumulative concentration-response curves to Phe in isolated rat corpus cavernosum from SD (■), SD + SP25 (☐), SD + SP50 (■), SD + SP100 (■) and SD + sild groups (■) (**A**) and HD (●), HD + SP25 (○), HD + SP50 (●), HD + SP100 (●) and HD + sild groups (●) (**B**). Symbols and vertical bars represent mean and S.E.M., respectively (*n* = 5). One-way ANOVA followed by Tukey’s post-test. * *p* < 0.05 (SD vs. SD + SP50 (A); SD vs. HD (B)); ^@^
*p* < 0.05 (SD + SP50 vs. SD + SP25, SD + SP100 and SD + Sild) and ^#^
*p* < 0.05 (HD vs. HD + SP25, HD + SP50, HD + SP100 and HD + Sild).

**Figure 6 marinedrugs-20-00467-f006:**
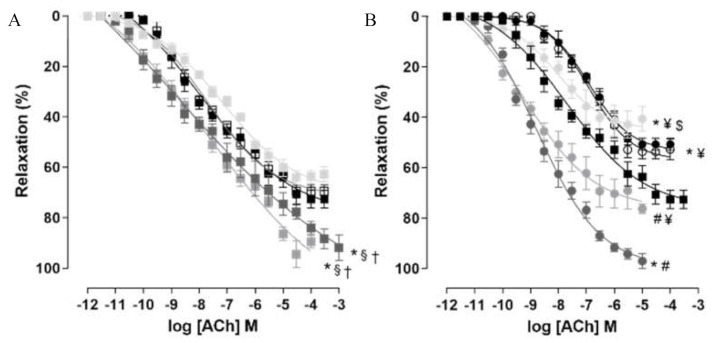
Cumulative concentration-response curves to ACh in isolated rat corpus cavernosum from the SD (■), SD + SP25 (☐), SD + SP50 (■), SD + SP100 (■) and SD + sild groups (■) (**A**) and HD (●), HD + SP25 (○), HD + SP50 (●), HD + SP100 (●) and HD + sild groups (●) (**B**). Symbols and vertical bars represent mean and S.E.M., respectively (*n* = 5). One-way ANOVA followed by Tukey’s post-test. * *p* < 0.05 (SD vs. SD + SP50 and SD + Sild (**A**); SD vs. HD, HD + SP25, HD + SP100 and HD + Sild (**B**)); ^§^
*p* < 0.05 (SD + SP25 vs. SD + SP50 and SD + Sild); ^†^
*p* < 0.05 (SD + SP100 vs. SD + SP50 and SD + Sild); ^#^
*p* < 0.05 (HD vs. HD + SP50 and HD + Sild); ^$^
*p* < 0.05 (HD + SP50 vs. HD + SP100) and ^¥^
*p* < 0.05 (HD + Sild vs. HD + SP25, HD + SP50 and HD + SP100).

**Table 1 marinedrugs-20-00467-t001:** Initial body mass, final body mass and body mass gain values of rats from the SD, SD + SP25, SD + SP50, SD + SP100 and SD + Sild groups.

Group	Initial Body Mass (g)	Final Body Mass (g)	Body Mass Gain (g)	Body Mass Gain (%)
SD	147.6 ± 8.4	322.9 ± 4.0	166.3 ± 10.5	118.2 ± 2.4
SD + SP25	168.2 ± 9.3	354.4 ± 14.3 ^@^	186.2 ± 10.0 ^@^	110.4 ± 1.5 ^@^
SD + SP50	157.4 ± 4.4	284.4 ± 10.1	129.1 ± 9.7	80.0 ± 3.4 *
SD + SP100	161.8 ± 7.8	338.6 ± 12.5 ^@^	176.8 ± 10.3 ^@^	109.0 ± 2.5 ^@^
SD + Sild	175.7 ± 2.6	358.8 ± 8.7 ^@^	183.1 ± 9.2 ^@^	104.5 ± 1.5 ^@^

One-way ANOVA followed by Tukey’s post-test. * *p* < 0.05 (SD vs. SD + SP50) and @ *p* < 0.05 (SD + SP50 vs. SD + SP25, SD + SP100 and SD + Sild) (*n* = 10).

**Table 2 marinedrugs-20-00467-t002:** Initial body mass, final body mass and body mass gain values of rats from the HD, HD + SP25, HD + SP50, HD + SP100 and HD + Sild groups.

Group	Initial Body Mass (g)	Final Body Mass (g)	Body Mass Gain (g)	Body Mass Gain (%)
SD	147.6 ± 8.4	322.9 ± 4.0	166.3 ± 10.5	118.2 ± 2.4
HD	158.1 ± 2.5	367.1 ± 12.8 *	215.0 ± 11.1 *	132.5 ± 3.0 *
HD + SP25	174.3 ± 17.2	320.7 ± 11.4 ^#^	148.6 ± 14.5 ^#^	84.0 ± 2.2 ^#^
HD + SP50	166.5 ± 5.3	310.2 ± 21.7 ^#^	145.3 ± 13.7 ^#^	86.0 ± 1.9 ^#^
HD + SP100	174.5 ± 10.5	308.6 ± 9.2 ^#^	143.1 ± 9.6 ^#^	77.0 ± 2.5 ^#^
HD + Sild	178.9 ± 4.0	344.4 ± 9.5	175.5 ± 6.2	92.0 ± 3.0

One-way ANOVA followed by Tukey’s post-test. * *p* < 0.05 (SD vs. HD) and ^#^
*p* < 0.05 (HD vs. HD + SP25, HD + SP50 and HD + SP100) (*n* = 10).

**Table 3 marinedrugs-20-00467-t003:** Biochemical parameters of rats from the SD, SD + SP25, SD + SP50, SD + SP100, SD + Sild, HD, HD + SP25, HD + SP50, HD + SP100 and HD + Sild groups.

Group	Glucose (mg/dL)	Triglyceride (mg/dL)	Total Cholesterol (mg/dL)	HDL-c (mg/dL)	LDL-c (mg/dL)
SD	86.9 ± 4.5	67.5 ± 2.5	53.2 ± 3.4	17.4 ± 2.4	14.9 ± 1.6
SD + SP25	88.1 ± 2.3	66.0 ± 6.0	44.3 ± 2.8	18.3 ± 2.1	14.4 ± 3.0
SD + SP50	86.7 ± 3.5	68.0 ± 7.2	48.0 ± 3.4	19.7 ± 1.7	15.5 ± 2.4
SD + SP100	78.8 ± 3.5	61.5 ± 4.7	49.8 ± 3.1	17.2 ± 1.3	18.2 ± 2.9
SD + Sild	84.3 ± 3.4	69.0 ± 4.1	49.8 ± 3.9	16.2 ± 1.1	15.9 ± 2.8
HD	91.4 ± 3.1	80.7 ± 7.2	53.0 ± 3.1	17.0 ± 1.8	14.3 ± 2.0
HD + SP25	86.2 ± 2.6	65.5 ± 6.4	41.3 ± 5.7	15.8 ± 2.0	12.9 ± 2.3
HD + SP50	89.7 ± 4.8	67.2 ± 4.8	48.3 ± 1.9	16.3 ± 1.0	13.5 ± 1.7
HD + SP100	87.2 ± 2.3	62.0 ± 3.4	50.7 ± 6.2	17.5 ± 1.1	14.0 ± 1.9
HD + Sild	85.6 ± 2.4	64.8 ± 8.9	47.3 ± 4.6	15.2 ± 1.1	13.0 ± 2.2

HDL-c = fraction of high-density lipoprotein; LDL-c = fraction of low-density lipoprotein (*n* = 10).

## Data Availability

Not applicable.

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
