# Peer review of "Supplementation with Spirulina platensis Prevents Damage to Rat Erections in a Model of Erectile Dysfunction Promoted by Hypercaloric Diet-Induced Obesity"

_marinedrugs, 2022, doi:10.3390/md20080467_

Round 1
Reviewer 1 Report
This is an interesting work. Anyway I have some consideration and suggestions for the authors.
Abstract
Is not so clear. Please, reformulate the abstract about groups and results obtained
Introduction
Pag 2 lines 46-48. I suggest to reformulate the text, in order of suggested treatments by guidelines as follows: lifestyle, pde5is, Testosterone replacement therapy ion case of hypogonadism, etc (see and use: PMID: 31196744, PMID: 29888533).
Material and methods
-I suggest to anticipate this section in the ms
-have you calculated the sample size of your population?
-please, explain why you have choosen sildenafil
Discussion
- this section is too extensive.
-Please, clarify better the strength and limits of your study at the end of this section.
Author Response
Dear reviewer,
The authors welcome comments and follow our updates on the article. Changes have been highlighted in the text.
Abstract
Is not so clear. Please, reformulate the abstract about groups and results obtained.
Answer: The authors reformulated part of the abstract to make it clearer, within the limits of the number of words allowed by the journal.
Introduction
Pag 2 lines 46-48. I suggest to reformulate the text, in order of suggested treatments by guidelines as follows: lifestyle, pde5is, Testosterone replacement therapy ion case of hypogonadism, etc (see and use: PMID: 31196744, PMID: 29888533).
Answer: The text has been redrafted as requested.
Material and methods
-I suggest to anticipate this section in the ms.
Answer: The section was placed in the sequence provided for the journal's rules.
-have you calculated the sample size of your population?
Answer: The sample followed the calculations from standardized in vivo or in vitro studies in our research laboratory. In this case, about 6-10 animals are used for the in vivo studies and about 5 animals are used for the in vitro studies.
-please, explain why you have choosen sildenafil.
Answer: Sildenafil was chosen because it is used as a standard drug in several studies on penile erectile activity.
Discussion
- this section is too extensive.
Answer: The authors have reduced part of the section to avoid reader fatigue.
-Please, clarify better the strength and limits of your study at the end of this section.
Answer: The authors tried to clarify the study's strength and limitations of the study.
Reviewer 2 Report
In this study, the authors evaluated the effect of a food supplementation with Spirulina platensis on body adiposity, oxidative stress, and erectile function of Wistar rats fed with differentiated diets, a standard and a hypercaloric diet, to verify whether improves penile function or prevents damage to erectile function due to increased caloric intake. The authors suggested that supplementation with S. platensis prevents damages associated to a hypercaloric diet consumption and emerges as an adjuvant for the erectile dysfunction (ED) prevention.
Comments
The reviewer has some concerns as follows:
- There are some confusing in the statistical analysis for data presentation. The indications of statistically significant symbols are not easy to understand, and the comparison among data is easy to be confused. The values with different letters for statistical significance according to the one-way analysis of variance (ANOVA) followed by Tukey’s post-test can be considered.
- In Table 3, why there are no statistical significance for blood glucose and lipids between HD and SD groups?
- In Figure 2A, did supplementation of S. platensis decrease the number of penile erections compared to SD alone group?
- In Figures 3 and 4, the “r” value is not high enough indicating that the correlation is not high, although “p” value is <0.05. It needs to be explained and discussed.
- In Figures 5 and 6, the data for contraction and relaxation are confusing. The data presentation is bad. Where is “A” or “B”?
- The Tables for presentation of Emax and pD2 from Figures 5 and 6 are suggested.
- The entire manuscript should be carefully checked for typos or inappropriate sentences.
Author Response
Dear reviewer,
The authors welcome comments and follow our updates on the article. Changes have been highlighted in the text.
There are some confusing in the statistical analysis for data presentation. The indications of statistically significant symbols are not easy to understand, and the comparison among data is easy to be confused. The values with different letters for statistical significance according to the one-way analysis of variance (ANOVA) followed by Tukey’s post-test can be considered.
Answer: The graphic data can be sent separately to the journal, for the preparation of the article in order to make the differences clearer for the readers..
In Table 3, why there are no statistical significance for blood glucose and lipids between HD and SD groups?
Answer: Statistical analyzes showed no difference between the values for the mentioned groups. The authors redone the analyzes and the difference was not identified by the statistical program.
In Figure 2A, did supplementation of S. platensis decrease the number of penile erections compared to SD alone group?
Answer: No, because the statistical analysis did not show any difference between the groups mentioned.
In Figures 3 and 4, the “r” value is not high enough indicating that the correlation is not high, although “p” value is <0.05. It needs to be explained and discussed.
Answer: The low correlation was highlighted in the discussion.
In Figures 5 and 6, the data for contraction and relaxation are confusing. The data presentation is bad. Where is “A” or “B”?
Answer: In the graph on the left (A), the data of the groups of animals that consumed the standard diet are observed. In the graph on the right (B), the data of the groups of animals that consumed the hypercaloric diet are observed. The symbols have been adjusted in the image.
The Tables for presentation of Emax and pD2 from Figures 5 and 6 are suggested.
Answer: The data were presented in the text, due to the maximum number of tables and graphs allowed by the journal's rules.
The entire manuscript should be carefully checked for typos or inappropriate sentences.
Answer: Answer: The text has been revised for better adjustments.
Round 2
Reviewer 1 Report
none
Reviewer 2 Report
This revised manuscript can be accepted. No further comments.